# Dysregulated Autophagy Mediates Sarcopenic Obesity and Its Complications via AMPK and PGC1α Signaling Pathways: Potential Involvement of Gut Dysbiosis as a Pathological Link

**DOI:** 10.3390/ijms21186887

**Published:** 2020-09-19

**Authors:** Ji Yeon Ryu, Hyung Muk Choi, Hyung-In Yang, Kyoung Soo Kim

**Affiliations:** 1Department of Clinical Pharmacology and Therapeutics, Kyung Hee University School of Medicine, Seoul 02447, Korea; rjy1532@khu.ac.kr (J.Y.R.); chl2813@khu.ac.kr (H.M.C.); 2East-West Bone & Joint Disease Research Institute, Kyung Hee University Hospital at Gangdong, Gandong-gu, Seoul 05278, Korea; yhira@khu.ac.kr

**Keywords:** sarcopenic obesity, AMPK signaling pathway, PGC-1α signaling pathway, aging, insulin resistance, inflammation, autophagy, gut axis

## Abstract

Sarcopenic obesity (SOB), which is closely related to being elderly as a feature of aging, is recently gaining attention because it is associated with many other age-related diseases that present as altered intercellular communication, dysregulated nutrient sensing, and mitochondrial dysfunction. Along with insulin resistance and inflammation as the core pathogenesis of SOB, autophagy has recently gained attention as a significant mechanism of muscle aging in SOB. Known as important cellular metabolic regulators, the AMP-activated protein kinase (AMPK) and the peroxisome proliferator-activated receptor-gamma coactivator-1 alpha (PGC-1α) signaling pathways play an important role in autophagy, inflammation, and insulin resistance, as well as mutual communication between skeletal muscle, adipose tissue, and the liver. Furthermore, AMPK and PGC-1α signaling pathways are implicated in the gut microbiome–muscle axis. In this review, we describe the pathological link between SOB and its associated complications such as metabolic, cardiovascular, and liver disease, falls and fractures, osteoarthritis, pulmonary disease, and mental health via dysregulated autophagy controlled by AMPK and/or PGC-1α signaling pathways. Here, we propose potential treatments for SOB by modulating autophagy activity and gut dysbiosis based on plausible pathological links.

## 1. Introduction

Population aging and the increasing incidence of obesity in recent years have been serious concerns worldwide [1,2,3]. Aging causes many physiological changes in the body, including important changes in body composition resulting in an increase in fat mass and visceral fat and a decrease in muscle mass and strength [4,5]. The decrease in muscle mass and strength and the increase in body fat proceed over a life time [6]. Increased fat and accumulation of lipids in non-adipose tissues have been demonstrated to be an evolutionary strategy in the process of human aging [7]. However, if the state of increased fat worsens, sarcopenic obesity (SOB) can develop, which is a harmful disease for the elderly. SOB is a new concept for obesity that is primarily age-related and is characterized by abnormal age-dependent muscle loss and concomitant fat accumulation [8]. Increased adiposity can be redistributed into the visceral organs and can infiltrate into bone and muscle [6]. Thus, visceral fat, intermuscular fat, intramyocellular fat, and ectopic organ fat are important factors for understanding SOB pathogenesis [9], and active prevention and management modalities are necessary due to the associated medical complications, risk for other age-related diseases, and general injury in the elderly.

Cellular disorders of metabolism are known to be key factors in SOB pathogenesis. Thus, autophagy activity, which removes dysfunctional organelles in aging cells, has been an important target for the treatment of SOB and to explain the molecular aspects of the disease. Appropriate regulation of the autophagic process is required for the maintenance of skeletal muscle mass by improving the quality of mitochondria [10]. The AMP-activated protein kinase (AMPK) and the peroxisome proliferator-activated receptor-gamma coactivator-1 alpha (PGC-1α) signaling pathways have been implicated in metabolic homeostasis and the activation of autophagy in tissues. These signaling pathways also seem to be involved in the gut–skeletal muscle axis. The gastrointestinal tract and the gut microbiome are thought to be deeply associated with diet, muscle function, and metabolism [11].

In this review, we focus on the important role of AMPK and PGC-1α signaling for autophagy activity. It is assumed that dysregulated signaling of the gut–skeletal muscle axis could induce SOB. Here, we propose a pathological link between SOB and its complications such as metabolic disease and cardiovascular disease, pulmonary disease, liver disease, falls and fractures, osteoarthritis, and mental health in the elderly. A deeper understanding of AMPK and PGC-1α signaling regulation in normalizing autophagy and gut dysbiosis could lead to proposed prospective therapeutic targets for SOB.

## 2. Age-Related SOB Is the Result of a Vicious Cycle between Aging, Sarcopenia, and Obesity

Aging, sarcopenia, and obesity are closely associated with each other from a pathophysiological point of view. Understanding the interrelationship among these factors is important for preventing and treating SOB. Aging can affect many functions of adipose tissue by changing the inflammatory mediators produced by adipocytes. Pre-adipocyte number and function and adipose tissue infiltration of macrophages and other lymphocytes can be modified by aging [12,13]. Aging causes the accumulation of adipose tissue in the abdominal cavity and the infiltration of adipose tissue into the muscle. Furthermore, ectopic fat deposition in the liver, the pancreas, or the heart can impair respective organ function and contributes to other age-related diseases [12,13,14]. Increased adipose tissue during aging leads to the production of adipo-cytokines (adipokines) such as tumor necrosis factor-α (TNF-α) and interleukin-6 (IL-6), which induce low grade chronic inflammation [15]. This condition is also referred to as “inflammaging” and “metaflammation”, the pathological concept important in the vicious cycle of aging and obesity. Inflammaging refers to low-grade pro-inflammatory state in which cytokines such as TNF-α and IL-6 increase during the aging process, while metaflammation refers to metabolic inflammation due to metabolic diseases caused by overnutrition [16,17]. This inflammation condition can alter metabolism in the process of age-related diseases such as sarcopenia [17].

Sarcopenia could be the initial step in obesity, or vice versa [18]. Sarcopenia induces decreased energy expenditure, physical inactivity, and hormonal changes and thus could cause obesity. Low muscle mass reduces basic metabolic rate, and muscle weakness may reduce physical activity. Physical inactivity and reduced basic metabolic rate cause the decline in total energy expenditure [18]. Meanwhile, obesity could accelerate sarcopenia. The obese state means increased total and intra-abdominal fat mass that can raise total adipocyte size and the number of infiltrated macrophages in adipose tissue. Macrophage secreted-adipokines such as TNF-α and IL-6 can induce low grade inflammation; thus, obesity could lead to insulin resistance (IR), increased leptin, and decreased adiponectin. Subsequently, IR causes impaired muscle protein turnover [19]. These factors eventually lead to the loss of muscle mass and muscle strength, which is called sarcopenia, which induces a vicious cycle [20,21,22,23,24].

As aging is associated with a gradual decline in muscle mass and strength, it can significantly contribute to sarcopenia. One of the underlying mechanisms is that aging leads to a reduction of many anabolic muscle signals by decreasing growth hormones such as insulin-like growth factor-1 (IGF-1) and estrogen. Another is an increase in catabolic signals with aging [25]. Aged muscle has high levels of myogenic repressor proteins such as Id-1, Id-2, and Id-3, along with markers of apoptosis such as BAX and caspase-9. A study using young and aged rats showed that aging activates apoptotic pathways through Id-induced BAX upregulation [26]. According to a large-scale analysis of the effects of age on murine behavior, aging increased body weight and reduced neuromuscular strength and motor dysfunction in older mice [27].

Obesity may accelerate aging, as obesity deteriorates cellular processes that are similar to aging. It has been reported that accumulated fat is a counterpart of aging [28]. Another study postulated the concept of “adipaging” to demonstrate the links between obesity and aging. Obesity and aging are involved in the deterioration of biological processes by IR, lipotoxicity, dysfunctional adipose tissue, oxidative stress, and immune dysfunction [29]. Thus, it is suggested that aging, sarcopenia, and obesity have a close triangular relationship where each can aggravate and exacerbate the others. The illustration of a “vicious triangle” indicates that the three factors could result in the development of age-related SOB by having adverse effects on each other throughout life (Figure 1).

## 3. Organokines Such as Myokines, Adipokines, and Hepatokines Are Involved in Modulating AMPK and PGC-1α Signaling Pathways, Which Are Important for the Induction of SOB

SOB is characterized by impaired crosstalk between skeletal muscle, adipose tissue, and the liver; therefore, a better understanding of these interactions is important to understand the disease. Muscle, adipose tissue, and the liver intercommunicate with each other similar to an endocrine organ through the use of organokines [30]. Myokines, adipokines, and hepatokines are the cytokines secreted from muscle, adipose tissue, and the liver, respectively. Mutual communication through various inter-related signaling pathways between these tissues may play an important role in regulating metabolic homeostasis, including myogenesis, adipogenesis, protein turnover, lipogenesis, and lipolysis [31]. Many different types of organokines could be associated with the modulation of AMPK and PGC-1α signaling pathways. The organokines include myostatin, myonectin, IL-15, and brain-derived neurotrophic factor (BDNF) as myokines, fibroblast growth factor 21 (FGF21) as an hepatokine, IL-6 and TNF-α as adipo-myokines, and adiponectin as an adipokine. As shown in Figure 2, they could be involved in modulating lipid metabolism, autophagy, and insulin sensitivity, leading to the induction of SOB.

Myostatin, the first known myokine, is critical for fat shift events and the inhibition of satellite cell activation and myoblast proliferation in muscle. Thus, it suppresses skeletal muscle hypertrophy through the downregulation of Akt/mTOR/p70S6K signaling [32]. Recently, it has been shown that myostatin can suppress protein synthesis through regulation of the AMPK signaling pathway [33], and decreased myostatin expression due to mutation causes the doubling of muscle mass as well as the browning of white adipose tissue [34]. Additionally, myostatin-knockout mice showed a striking increase in insulin sensitivity and a reduction in fat mass [35]. Furthermore, myostatin may be involved in maintaining basal levels of autophagy [35,36]. Meanwhile, myonectin plays an important role in increasing muscle mass and, in contrast, has the ability to suppress autophagy activity [37] and increase glucose uptake and fatty acid oxidation through activation of the AMPK signaling pathway in rat skeletal myocytes [38]. IL-15, one myokine, also has anabolic effects in the muscle–fat interaction through the modulation of body composition and insulin sensitivity [39]. IL-15 reduces adipose mass and also induces mitochondrial activity in adipose tissue [40,41,42,43], and IL-15 mRNA level and plasma concentration are decreased by defective AMPK signaling, which indicates that IL-15 may function through the AMPK signaling pathway. In aged muscle, it was demonstrated that aging could impair IL-15 signaling with aging-induced defective AMPK activation [44,45].

BDNF could contribute to mediate muscle satellite cell and fatty acid oxidation [46]. An in vivo and an ex vivo study indicated that BDNF increased the phosphorylation of AMPK [47]. Electrically stimulated C2C12 cells also showed increased BDNF mRNA and protein levels through AMPK activation. Conversely, chronic secretion of IL-6 and TNF-α, which are adipo-myokines, augments fat mass and reduces muscle mass by exerting influence on IR, energy metabolism, and the secretion of growth hormones [31]. Furthermore, the InCHANTI study, a representative population-based study of older persons living in the Chianti geographic area (Tuscany, Italy), indicated that SOB was linked to elevated levels of IL-6 [48]. Meanwhile, the beneficial effect of IL-6 increases lipolysis and fatty acid oxidation in adipose tissue [49]. It also improves glucose uptake and fatty acid oxidation as well as increases insulin-stimulated glucose disposal through AMPK signaling [50]. Studies of the effect of IL-6 on skeletal muscle insulin action have shown that exposure time induces differential effects of IL-6. For example, IL-6 activates AMPK signaling during short-term exposure to increase insulin sensitivity, and chronic IL-6 exposure triggers insulin resistance [51]. Therefore, IL-6 has dual harmful and beneficial effects according to long-term or short-term expression. TNF-α also inhibits myoblast differentiation and increases intramuscular diacylglycerol accumulation, causing IR in skeletal muscle [52]. TNF-α develops IR in adipose tissue and activates angiogenesis and regenesis in adipose-derived stem cells [53]. IL-6 and TNF-α tend to suppress the AMPK signaling pathway [54]. Adiponectin, one of the major adipokines, plays a significant role in energy homeostasis and the insulin signaling pathway. It also improves fatty acid oxidation in skeletal muscle by increasing AMPK phosphorylation and increases mitochondrial oxidative metabolism and biogenesis [55,56]. A recent study showed that adiponectin could induce autophagy and reduce oxidative stress under pathological conditions of a high-fat diet in mice [57]. Adiponectin also mediates fatty acid oxidation in skeletal muscle through the AMPK signaling pathway [58].

PGC-1α expression is also involved in organokine activity. For example, the expression of PGC-1α in skeletal muscle can stimulate fibronectin type 3 domain-containing protein expression (FNDC5), and irisin is a cleaved form of FNDC5 [38,59]. Irisin improves the growth and the differentiation of muscle fibers and accelerates thermogenesis through fat browning in adipose tissue [60]. It is also known that irisin improves glucose homeostasis, increases adipocyte energy expenditure, and modulates the expression of metabolic enzymes and intermediates [31,61]. FGF21, secreted from the liver, has a direct effect on glucose homeostasis, lipid metabolism, and insulin sensitivity in skeletal muscle [62,63]. It stimulates not only PGC-1α function but also AMPK signaling pathways by several direct and indirect ways [64]. One study showed that FGF21 expression is promoted by autophagy deficiency and its subsequent mitochondrial dysfunction [65].

In conclusion, AMPK and PGC-1α signaling pathways are regarded as key regulators of metabolic homeostasis. They are important signaling pathways in response to organokines, which is significant for the intercommunication between skeletal muscle, adipose tissue, and the liver. Among them, FGF21, myonectin, and adiponectin affect the AMPK and the PGC-1α signaling pathways, which then have an effect on autophagy. The fact that autophagic activity is largely regulated by the AMPK and the PGC-1α signaling pathways is helpful for understanding SOB and the pathological link with its complications.

## 4. Dysregulated Autophagy Aggravates Sarcopenic Obesity through Dysfunctional AMPK and PGC-1α Signaling Pathway-Mediated Inflammation and Insulin Resistance

### 4.1. Role of Autophagy in the Induction of Sarcopenic Obesity

Autophagy is an intracellular degradation process that removes dysfunctional organelles and denatures proteins in living cells [66]. It has been demonstrated that the overweight condition in elderly people induces impaired autophagy in skeletal muscle [67]. Autophagy plays an important role in SOB during the aging process as a main regulator of insulin homeostasis and metabolic homeostasis [68]. The regulatory mechanisms of autophagy activity can be regulated in response to various metabolic stimuli [69]; thus, impaired autophagic signaling pathways in aging contribute to a buildup of intracellular aggregates, leading to disturbed cellular and tissue homeostasis and functional loss [70]. Adequate induction and normal regulation of the autophagic process is required for the maintenance of skeletal muscle mass and strength. The activation of autophagy seems to suppress skeletal muscle loss through variable signaling pathways, i.e., AMPK, PI3K-mTOR, and PGC-1α [10,71]. It seems that AMPK activators and mTOR inhibitors could modulate autophagy activity; additionally, autophagy activation can suppress inflammasome activation, which elevates low-grade inflammation and the aging process [72].

AMPK activation is significant in autophagy activation signaling. Thus, AMPK controls both protein synthesis and protein degradation through an autophagic process [10]. In response to nutrient deficiency and exercise, AMPK increases autophagy activity by activating FoxO3, which is an important transcription factor for human longevity and inhibiting the mTOR pathway [73]. AMPK activates FoxO3 by regulating the transcription of autophagy-related (ATG) proteins [74]. Bcl2 is also involved in autophagy activation. Bcl-2 is an antiapoptotic protein that binds to Beclin1 during non-starvation status and inhibits autophagy; however, Bcl-2 dissociation from Beclin1 activates autophagy. Activated AMPK can also phosphorylate JNK1 and thus initiate the JNK1-Bcl-2 pathway to dissociate the Beclin1-Bcl-2 complex to evoke autophagy [75,76]. Moreover, PI3K-Akt-mTOR signaling pathways have received attention to understand the pathogenesis of SOB. As a central pathway in muscle metabolism, PI3K-Akt-mTOR pathways are activated by the binding of insulin and insulin-like growth factor 1 (IGF-1) to membrane receptors. The mTOR pathway plays a negative role in the initiation of autophagy through the phosphorylation of the ULK1 complex in skeletal muscle. Furthermore, aged rats showed that the expression of Atg proteins such as Atg5, Atg7, and LC3 protein was lower than that in young rats. This suggests that aging could decrease autophagy activity as a result of the reduced function of skeletal muscle by accumulating damaged organelles such as mitochondria [75,77].

### 4.2. Dysregulated AMPK and PGC-1α Signaling Pathways Contribute to Inflammation and Insulin Resistance

Disturbed inflammatory signaling pathways and insulin signaling pathways are found in sarcopenic muscle [78]. Aging, IR, and inflammation have an effect on autophagy activity via dysregulation of AMPK and PGC-1α signaling. First, AMPK and PGC-1α activation tend to decrease with aging [79]. A study showed that the active state of AMPKα was decreased, but its expression did not change in old rats, suggesting aging impairs AMPK activation but not its expression [80]. Meanwhile, both PGC-1α expression and activation were significantly decreased in the aged groups [81]. Second, both suppression of AMPK and PGC-1α signaling and elevation of PI3K-Akt-mTOR signaling could decrease autophagy activity, which leads to IR [82]. Not only aging but also declined autophagy activity leads to an increase in the risk of IR by decreasing GLUT4 protein level in old rats [80]. A study indicated that insulin-resistant myocytes increased the expression of some markers showing decreased autophagy activity, which implies declined autophagic activity of the skeletal muscle [83]. Blocking autophagy activity with the lysosomal inhibitor chloroquine diminished insulin sensitivity, while increasing autophagic activity improved insulin sensitivity in myotubes [83]. A study of autophagy in diabetes also indirectly indicated that decreased autophagy activity induced IR [84]. IR can suppress AMPK signaling, thus generating a vicious cycle between AMPK, autophagy, and insulin resistance [69]. Third, decreased autophagy activity with aging aggravates inflammation through dysregulated AMPK signaling pathways. The increased inflammation decreases activation of the AMPK signaling pathway and vice versa [72]. Additionally, decreased autophagy activity accumulates dysfunctional mitochondria or impaired intercellular organelles, which could increase reactive oxygen species (ROS) production. The ROS production could stimulate intracellular danger-sensing multiprotein platforms called inflammasomes, which induce inflammation [85,86]. For example, the inflammasome is induced by multiple cellular stresses. Nod-like receptor 3 (NLRP3), which is one of inflammasome members, can be triggered by increased levels of ROS [87,88]. Furthermore, NF-κB signaling also stimulates NLRP3 activation [89]. Activated NF-κB signaling is one of the critical signaling pathways in SOB and in the aging process in mice [90]. Thus, the inhibition of autophagic activity in SOB could generate an inflammatory reaction through the activation of inflammasomes, in particular, NLRP3 [72]. Therefore, it is concluded that dysregulated autophagic activity, inflammation, and IR play pivotal roles in the induction of SOB and its complications.

### 4.3. Endoplasmic Reticulum Stress Disturbs the Function of Autophagy and Dysfunctional Autophagy Also Inhibits ER Function

As shown in Figure 3, endoplasmic reticulum (ER) stress is one of the important factors causing uncontrolled autophagy, inflammation, and IR in SOB. It also suppresses AMPK and PGC-1α signaling pathways [91]. Reversely, the activation of AMPK signaling can alleviate ER stress [92]. In the normal state, ER stress could trigger autophagy activity through several pathways such as IRE1-JNK signaling. Another ER-stress related factor, PERK, phosphorylates eIF2α and induces autophagy activity. Increased ER stress could downregulate the Akt-mTOR signaling pathway, which stimulates autophagy activity. However, the above pathways are impaired in SOB, and the accurate mechanism is not yet revealed. Declined autophagy activity in SOB could not alleviate ER stress, which leads to chronic ER stress and inflammation. Chronic ER stress aggravates SOB through the downregulation of key proteins required for autophagy activity [91]. In conclusion, these interactions form a vicious cycle between chronic ER stress and impaired autophagic activity, as shown in Figure 3. ER stress is considered a crucial element that induces IR in several ways; it deteriorates IR by impairing the IRE1-JNK pathway in muscle cells [93] and also disturbs insulin signaling by promoting fat accumulation, which aggravates inflammation [91].

## 5. Sarcopenic Obesity Increases the Risk of Various Diseases

It is recognized that SOB is clinically associated with other diseases [14,94,95]. Based on a meta-analysis, SOB is a significant predictor of all-cause mortality among older patients [96]. It is deeply connected to age-related diseases as well as systemic inflammation and dysfunction of organs. Clinical studies demonstrated the association between SOB and metabolic disease, dysmobility syndrome, mental diseases, lung diseases, renal diseases, osteoarthritis, and digestive diseases (Figure 4). Understanding the association of AMPK and PGC-1α signaling pathways with autophagy activity could provide new insight to elucidating the pathologic link with other diseases.

### 5.1. Association of SOB with Metabolic Disease and Cardiovascular Disease

SOB in the elderly is closely associated with increased risk of metabolic and cardiovascular diseases such as atherosclerosis, type 2 diabetes, and cardiac hypertrophy. Atherosclerosis, one of the major causes of cardiovascular disease, is characterized by excessive lipid accumulation with endothelial cell dysfunction [97]. AMPK and autophagy are key factors to alleviate and regulate atherosclerosis combined with SOB [98]. AMPK activation inhibits acetyl-CoA carboxylase (ACC) for fatty acid synthesis and inhibits 3-hydroxy-3-methylglutaryl coenzyme A reductase (HMGR) for sterol synthesis with fatty acid oxidation and glycolysis decomposition. Furthermore, AMPK-induced autophagic activity stimulates ATP-binding cassette transporter A1 expression, causing cholesterol efflux [99]. With age-related SOB, AMPK signaling and autophagy activity could be decreased such that the risk of atherosclerosis is higher with age-related SOB. Additionally, several studies showed NLRP3 inflammasome activation could accelerate the pathogenesis of atherosclerosis, type-2 diabetes, and other age-related diseases [72]. The inflammasome is involved in declined autophagy and inflammation in SOB. One study demonstrated that sarcopenic obese mice had interrupted hypertrophic regulating pathways, which control the enlargement and the growth of cells [52]. These mice showed increased risk of cardiac hypertrophy and decreased autophagy activity. [52]. Additionally, protein level of PGC-1α in vessels from patients with atherosclerosis is decreased compared to healthy groups [100,101].

In a clinical study, the SOB group had a close association with IR, metabolic syndrome, and cardiovascular disease risk factors relative to any other group in the elderly population [102]. The relationship between body composition in SOB patients and a variety of cardiometabolic risk factors, including blood pressure, glucose tolerance indices, lipid profiles, inflammatory markers, and vitamin D level, was demonstrated by a study among 2945 subjects 60 years of age or older. In a study with 6832 adults who participated in the 2009 Korea National Health and Nutrition Examination Survey, the SOB group had higher systolic and diastolic blood pressure than those in the normal group [103]. This study also indicated that abdominal obesity and sarcopenia may act synergistically to induce hypertension. Another study showed that patients with SOB have a higher risk of metabolic syndrome, type 2 diabetes, and atherosclerosis [104]. According to a recent systemic review including 11 articles about the prevalence of SOB and the prevalence of diabetes, co-existence of sarcopenia and obesity increases the risk of type 2 diabetes by nearly 38% when compared with those with obesity alone [105]. The five-year study with 1231 men aged over 70 years demonstrated that older men with SOB have increased probability of common metabolic syndrome compared with counterparts with healthy body habitus [106]. In addition, strong theoretical reasons indicated SOB to be a predictor of cardiovascular disease [107]. A prospective cohort study with participants including 3366 men and women older than 65 years found that cardiovascular disease risk was significantly increased by 23% within the sarcopenic-obese group [108]. In a study with 3320 Korean adults 40 years of age or older, the results indicated that SOB was closely associated with increased cardiovascular disease risk in Korean adults [109].

### 5.2. Association of SOB with Liver Disease

Nonalcoholic fatty liver disease (NAFLD) and nonalcoholic steatohepatitis (NASH) account for most liver diseases. Clinical research indicates that sarcopenia and obesity are associated with liver disorders. NAFLD is characterized by excessive accumulation of hepatic lipids and is closely related to decreased AMPK activation [110]. AMPK regulates energy homeostasis by stimulating lipid metabolism through inhibition of ACC 1 and 2, which induce lipid synthesis. More specifically, AMPK inhibits the conversion of acetyl-CoA to malonyl-CoA by phosphorylating ACC1 at Ser79 and ACC2 at Ser212. Because malony-CoA is a precursor in fatty acid synthesis and an allosteric inhibitor of fatty acid oxidation, it means that AMPK can stimulate lipid metabolism [111]. Thus, genetically induced liver specific AMPK activation in mice protected against NAFLD by inhibiting de novo lipogenesis and promoting fatty acid oxidation [112]. This study also revealed that AMPK activation can reduce lipid accumulation in the liver and protect the liver from inflammation and fibrosis. As with AMPK, PGC-1α is a pivotal factor in the metabolic regulation of liver disease. Overexpression of hepatic PGC-1α reduced hepatic triacylglycerol (TAG) storage and serum TAG levels via adenoviral transduction of PGC-1α in rats [113]. A human study showed that PGC-1α expression was lower in the liver of the obese group than the lean group [114]. Additionally, mitochondrial autophagy can promote mitochondrial fatty acid oxidation, and it can inhibit hepatic fatty acid accumulation and improve hepatic IR [115].

Clinical studies also support the relationship between SOB and liver disease. SOB seems to share a similar pathological background with IR and inflammation in liver disease. A study of 309 patients with sarcopenia showed that sarcopenia is significantly associated with NASH and fibrosis and is independent of obesity, inflammation, and IR [116]. In contrast, another study suggested that there was an independent association between sarcopenia and NAFLD [117]. Another meta-analysis study demonstrated that sarcopenia is not only a risk factor for NAFLD but is also associated with the progression of NAFLD-related significant fibrosis [118]. Decreased muscle mass combined with increased visceral fat mass worsens the hepatic conditions of NAFLD [119]. It was concluded that SOB is a risk factor for the development of fatty liver and may accelerate liver fibrosis in patients with fatty liver [120].

### 5.3. Association of SOB with Falls and Fractures and Osteoarthritis

Dysmobility syndrome has been described as a new approach to help identify older people at risk of poor health outcomes [121,122,123,124,125]. There are several studies indicating that impaired muscle function and bad body composition have significant implications for falls and fractures in older men. According to a study of 1486 older men from the Concord Health and Aging in Men Project (CHAMP), EWGSOP-defined SOB men are connected with increased fall rates over two years, while FNIH-defined SOB men have increased fracture risk over six years compared with non-SOB men [126]. Another study including 1089 community-dwelling older adults showed that sarcopenic obese older men had a three-fold increased rate of self-reported fractures over 10 years compared to both non-sarcopenic non-obese and obese alone counterparts [127]. In addition, sarcopenic obese older adults have an increased risk of osteoporosis and incident non-vertebral fracture [127]. Moreover, a cohort study of 451 elderly men and women followed for up to eight years concluded that the SOB group was 2.63 times more likely to report instrumental activities of daily living disability than lean sarcopenic or non-sarcopenic obese subjects and those with normal body composition [24].

Osteoarthritis (OA), which is characterized by progressive deterioration of the articular cartilage and joint disability, is clinically related to SOB. As with SOB, decreased AMPK activity was detected in OA cartilage [128]. Accelerated OA progression was displayed in AMPK conditional knockout mice, indicating AMPK activity is important in articular cartilage and joint homeostasis. AMPK activation in chondrocytes was shown to induce anti-catabolic effects in vitro [129]. The pro-catabolic response in chondrocytes from patients with OA can result from a decrease in PGC-1α activity modulated by AMPK deficiency [130]. In addition, autophagy is known to have chondroprotective effects but is defective in OA patient chondrocytes [131]. Clinical studies showed that SOB and OA are associated with each other along with impaired physical function. A cross-sectional clinical study including patients with obesity and knee OA demonstrated that SOB with end-stage knee OA influenced physical function, strength, and aspects of quality of life with slower walking speed, lower walking endurance, and lower grip strength [132]. Additionally, SOB may have an effect on therapeutic and surgical outcomes in knee and hip OA treatments [133]. A cross-sectional study also concluded that SOB is more closely associated with knee OA than non-SOB patients [134].

### 5.4. Association of SOB with Pulmonary Disease and Mental Health

SOB can negatively affect the lungs and mental health. AMPK is considered an important regulator of metabolic recovery in the lungs in a mouse model, but its function is impaired by aging [135]. PGC-1α levels have been shown to be repressed in idiopathic pulmonary fibrosis [136], and another study found that there is a negative effect of low muscle mass along with obesity on lung damage in elderly patients. This means that, as aging proceeds, the change in body composition is a risk factor for lung function [137].

SOB could also be a risk factor for depressive symptoms. Clinical research using data from 3862 community dwelling adults in the English Longitudinal Study of Aging showed that a reduction in grip strength over four years was associated with a higher risk of depressive symptoms in only obese participants [138]. In other aspects, decreased AMPK and PGC-1α signaling pathway activity seem to be related to depression as well as SOB. A study using mice exposed to chronic stress showed that depression-like behaviors are associated with inactivation or decreased phosphorylation of AMPK. In addition, PGC-1α mRNA levels were lower in patients with depression symptoms compared with healthy controls [139].

## 6. Potential Involvement of Gut Dysbiosis in the Pathological Link between Sarcopenic Obesity and Various Diseases

The gut microbiota is a hot topic in medicine in recent times, and it seems that the gut microbiota is closely associated with skeletal muscle in SOB pathogenesis and inter-organ communication (Figure 5). Gut dysbiosis can be interpreted as a disturbance in the composition of the gut microbiota, which can mean an increase in pathogenic species and a loss of beneficial species for humans as well as a loss of species variety [140], and a senescent gut can cause trouble in the digestive process. Gut dysbiosis caused by aging, diet, lifestyle, or disease disturbs normal communication between the gut and other organs such as the brain, the liver, and the muscle. Gut dysbiosis is associated with complications of SOB such as atherosclerosis, NAFLD, and knee and lung diseases. First, it contributes to the progression of atherosclerosis through overwhelming reverse cholesterol transport and producing foam cells that deposit in arterial plaque [141]. Second, the liver and the gastrointestinal tract have a bidirectional relationship through the biliary duct, which is named the gut–liver axis. Thus, it has been suggested in many studies that gut dysbiosis plays a role in the pathogenesis of NAFLD. Decreased *Firmicutes/Bacteroidetes* ratio can induce unfavorable inflammatory pathways and metabolic pathways such as short chain fatty acids (SCFAs)-induced de novo lipogenesis, increased lipopolysaccharide (LPS)-induced liver inflammation, and altered bile acid profile, which consequently result in NAFLD [142]. Third, a possible correlation between gut dysbiosis and the progression of OA has been reported. A pro-inflammatory microbial shift in the intestine accelerated cartilage degeneration through increased macrophage presence in the knee and upregulation of monocyte chemokine MCP-1 in synovial tissue [143]. Changes in gut microbial composition seemed to increase intestinal permeability that induced LPS translocation, provoking chronic low-grade inflammation and structural damage in the knee joint in a rat experiment [144]. Fourth, the bidirectional connection between the gut and lung could be a possible pathologic link between SOB and lung impairment [145].

Analysis of the gut microbiome is important to elucidate the mechanism by which the intestinal microbiome is involved in the link between SOB and its complications. Because there are no studies on the association between the human gut microbiome and SOB, such studies would be helpful to gain insight into treatment methods. As humans age, the incidence of comorbidities associated with intestinal microflora increases [146]. The reason is presumed to be that the gut microbiota and the gastrointestinal tract are deeply involved in metabolism and inflammation through direct and indirect pathways. However, the exact pathway linking the gut with other organs has not been well revealed. The main bacterial taxa such as *Bifidobacteria, Lactobacillus, Escherichia coli,* and *Faecalibacterium prausnitzii* are known to be implicated in the gut–skeletal muscle axis [147]. Some studies show that the elderly tend to have lower levels of *Bifidobacterium* and *Lactobacillus*, known as anti-inflammatory bacteria, and decreased levels of *Faecalibacterium prausnitzii* [148,149]. In addition, microbial metabolites from these bacteria are known to regulate muscle anabolism, for example, by increasing amino acid biosynthesis or AMPK signaling, or by decreasing oxidative stress. It seems that beneficial bacteria for muscle health are negatively associated with age in the senescent gut. Thus, age-related alterations of the gut microbiome composition could deteriorate muscle function and muscle protein synthesis. With respect to cell signaling, some studies suggested that the gut microbiome induces potential signaling pathways affecting the gut and the skeletal muscle interaction. For example, AMPK and PGC-1α signaling pathways seem to be involved in the gut–skeletal muscle axis [150]. In addition, PI3K-Akt-mTOR and myostatin/activin signaling pathways may be implicated in the axis by suppressing the NF-κB and FoxO signaling pathway [147]. However, further studies on humans and animals are required to elucidate more about the gut–skeletal muscle axis mechanism and its relationship to SOB.

In a *Drosophila melanogaster* experiment, the gut microbiota was also shown to communicate with inflammatory and oxidative stress pathways [151]. Altered gut microbiota diversity and weakened gut barrier integrity induce abnormal leakage of LPS, histamines, and serotonins, thereby instigating a high inflammatory response [146]. The LPS-activated toll-like receptor (TLR) 4 signaling pathway can mediate autophagy activation in muscle cells, which indicates that LPS could induce excessive autophagy activity, instilling another state of dysregulated autophagy and thus aggravating SOB. There may be interplay pathways between gut microbiota and muscle autophagy. A study focusing on intestinal diseases, especially inflammatory bowel disease, provided a perspective of interplay between autophagy and the gut microbiome. For example, defective autophagy activity could cause intestinal dysbiosis, resulting in ER stress and an inflammatory response [152].

However, it is controversial to say that the gut microbiome affects autophagy activity. Decreased muscle and strength in age-related SOB can be associated with excessive autophagy activity as well as deficient autophagy activity. Deficient autophagy activity is closely related with aging and cell senescence resulting in loss of the ability to remove dysfunctional organelles, while excessive autophagy activity may be associated with gut dysbiosis. We propose the potential involvement of gut dysbiosis in the pathological link between SOB and other various diseases. From understanding the interplay of AMPK and PGC-1α signaling pathways, autophagy, and gut dysbiosis, we can obtain new insights for SOB treatment (Figure 6).

## 7. Conclusions

The present review analyzes the potential mechanisms for how SOB heightens the risk of its complications. It seems that dysregulated autophagy activity, AMPK and PGC-1α signaling, and gut dysbiosis are greatly implicated in the pathological link between SOB and its complications. Crosstalk between organs via myokine, adipokine, hepatokine, or microbial metabolites has significant influence on inflammation and IR. Targeting dysregulated AMPK and PGC-1α signaling and autophagy activity and the gut microbiome could be a promising strategy to treat SOB and its complications. For example, the modulation of autophagy activity has been tried as one treatment against obesity [153]. Metformin improves NAFLD via AMPK-dependent activation of ACC in preclinical studies, resveratrol ameliorates hepatic steatosis by inducing autophagy activity via the AMPK signaling pathway [111,154], spermidine and rapamycin have autophagy-inducing effects in rodent models [155,156], and probiotics, prebiotics, and phytochemicals such as flavonoids and phenolic compounds are known to have an impact on improving gut dysbiosis in obesity [157,158].

Thus, further understanding the interplay of AMPK and PGC-1α signaling pathways, autophagy, and gut dysbiosis may lead to the development of effective novel therapies in SOB and its complications in elderly patients.

## Figures and Tables

**Figure 1 ijms-21-06887-f001:**
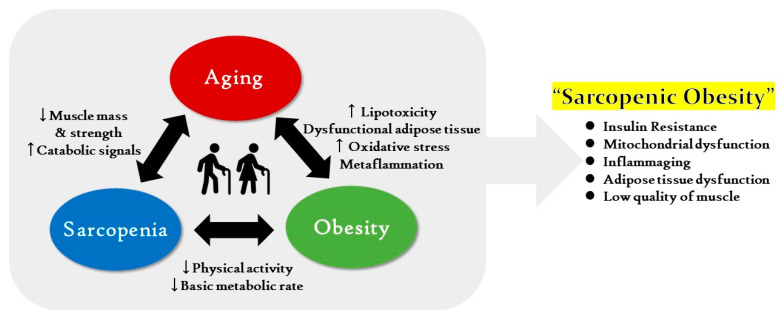
The “vicious-triangle” cycle between aging, sarcopenia, and obesity could cause sarcopenic obesity. The elderly are prone to be negatively affected by a vicious cycle between aging, sarcopenia, and obesity. Aging and sarcopenia are associated with low muscle mass and strength with increased catabolic signals. Aging and obesity have a close relationship via high lipotoxicity and oxidative stress with dysfunctional adipose tissue and metaflammation. Sarcopenia and obesity can aggravate each other by decreased basic metabolic rate with low physical activity. These associations could be an accelerator to induce the sarcopenic obese state, which is characterized by high fat mass and low quality muscle, insulin resistance (IR), and inflammaging along with mitochondrial dysfunction and muscle and adipose tissue dysfunction.

**Figure 2 ijms-21-06887-f002:**
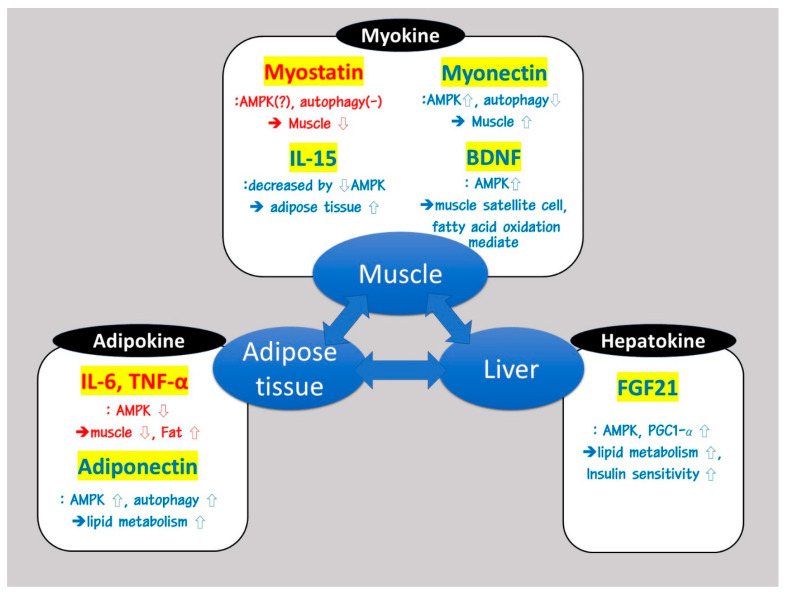
Role of myokine, adipokine, and hepatokine organokines on the induction of sarcopenic obesity (SOB). Organokines are significant for the intercommunication between skeletal muscle, adipose tissue, and liver. They modulate the AMP-activated protein kinase (AMPK) and the peroxisome proliferator-activated receptor-gamma coactivator-1 alpha (PGC-1α) signaling pathways, which are involved in lipid metabolism, autophagy, and insulin sensitivity, leading to the induction of SOB. Organokines are classified into myokines, adipokines, and hepatokines. Red color indicates a relatively harmful effect, while blue color indicates a relatively beneficial effect on metabolism.

**Figure 3 ijms-21-06887-f003:**
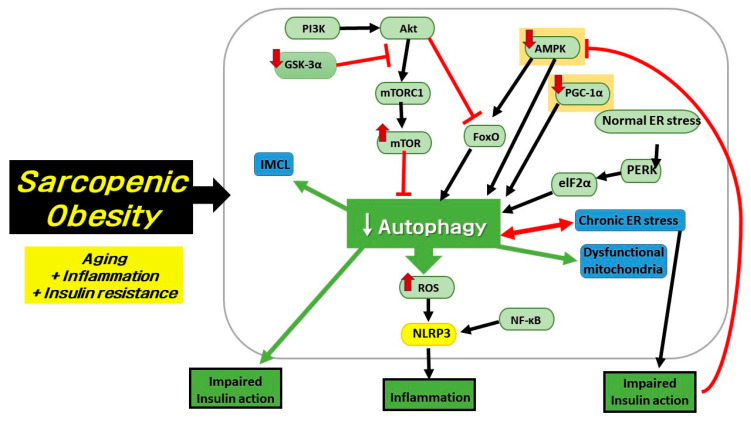
Important signaling pathways are related to deficient autophagy that causes sarcopenic obesity. Deficient autophagy is induced by several upstream signaling pathways to influence downstream cellular physiological phenomena. Suppressed activation of AMPK and PGC-1α with overactivated mTOR signaling through elevated PI3K-Akt signaling are the main regulatory pathways in autophagy failure related to aging, inflammation, and impaired insulin sensitivity. GSK-3α, which can inhibit hyperactivation of mTORC1, is lower in sarcopenic muscle. Deficient autophagy could induce high levels of reactive oxygen species (ROS) with dysfunctional mitochondria. The increase of ROS level is linked to inflammation by stimulating the inflammasome (NLRP3), which is also stimulated by NF-κB signaling. Deficient autophagy could result in higher intramyocellular lipid deposition in skeletal muscle. Chronic endoplasmic reticulum (ER) stress induces impaired autophagy, and vice versa. It is also another key factor to impair insulin action. Impaired insulin action, leading to IR, can inhibit AMPK activation. Thus, autophagy dysfunction deteriorates. Red lines indicate inhibition and black lines indicate stimulation. White and red arrow (↓, ↑) indicates decreased or increased activity, respectively.

**Figure 4 ijms-21-06887-f004:**
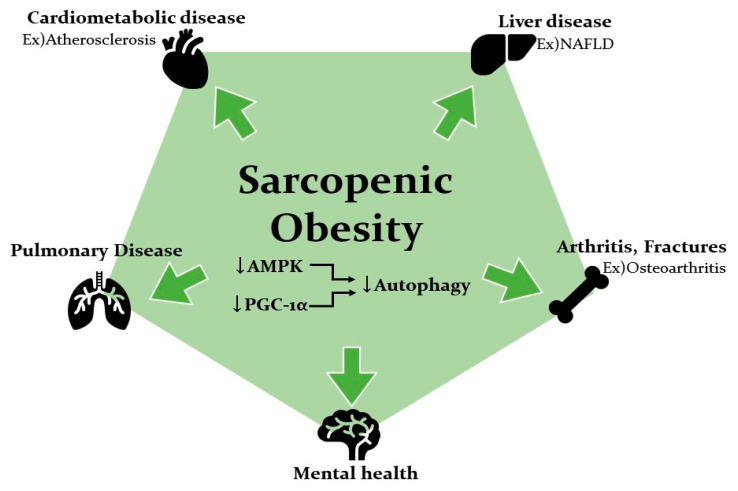
Pathological link between sarcopenic obesity and its complications. SOB seems to increase the risk of cardiometabolic disease, liver disease, arthritis, and lower lung function in addition to a negative effect on mental health. Low activation of AMPK and PGC-1α and the dysregulation of autophagy are proposed to play important roles in the pathological link between SOB and its complications.

**Figure 5 ijms-21-06887-f005:**
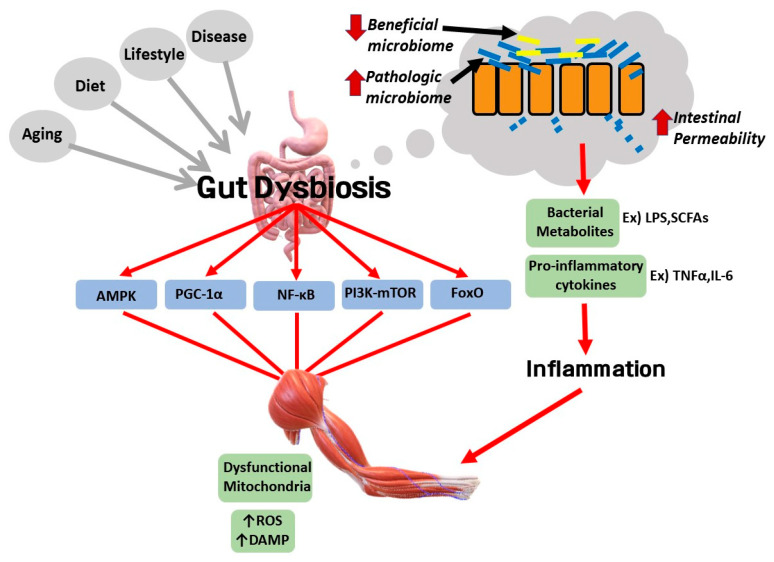
Potential pathway of the gut–skeletal muscle axis in sarcopenic obesity. Gut dysbiosis seems to negatively influence skeletal muscle health. AMPK, PGC-1α, NF-κB, PI3k-mTOR, and FoxO signaling are considered as potential signaling pathways in the gut–skeletal muscle axis. Gut dysbiosis can be triggered by aging, diet, lifestyle, and disease. The intestinal gut could lose the tight junction barrier and increase intestinal permeability with a less beneficial microbiome and higher pathologic microbiome. Toxic bacterial metabolites such as lipopolysaccharide (LPS) or single chain fatty acids (SCFAs) are disclosed through a leaky gut, which can trigger a pro-inflammatory response via inflammatory cytokines. Skeletal muscle could lose function by gut dysbiosis-induced inflammation and dysfunctional mitochondria. Additionally, ROS would increase and produce many DAMPs in skeletal muscle cells, aggravating SOB. Arrow (↓, ↑) indicates decreased or increased status, respectively.

**Figure 6 ijms-21-06887-f006:**
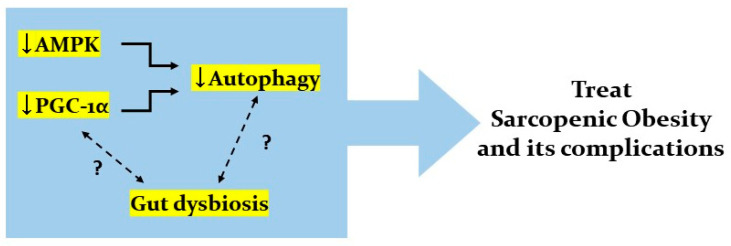
Possible interplay of AMPK and PGC-1α signaling pathways, autophagy, and gut dysbiosis. There could be possible interplay between AMPK and PGC-1α signaling pathways, autophagy, and gut dysbiosis. Dysregulated autophagy is one of the important targets to treat SOB and its complications. Decreased AMPK and PGC-1α signaling results in decreased autophagy. Gut dysbiosis and the gut–organ axis could regulate AMPK and PGC-1α signaling. Additionally, gut dysbiosis is closely related with excessive autophagy. Elucidating this interplay could be a key to new treatment modalities for SOB and its complications. Arrow (↓) and question marker (?) indicates decreased activity and undefined mechanisms, respectively.

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
