# Peer review of "Dysregulated Autophagy Mediates Sarcopenic Obesity and Its Complications via AMPK and PGC1α Signaling Pathways: Potential Involvement of Gut Dysbiosis as a Pathological Link"

_ijms, 2020, doi:10.3390/ijms21186887_

Round 1
Reviewer 1 Report
Here, the authors reviewed the link between sarcopenic obesity (SOB) and a number of systemic/organ diseases. Their work is well organized, and focuses on novel aspects involving gut microbiota. Figures are appropriate and help the readers understand the key message of the review.
My only major concern is the total absence of any mention to "inflammaging" and "metaflammation" as cardinal manifestations of the aging phenotype. In particular, the authors are strongly encouraged to revise paragraph #2 introducing these concepts with proper citations to seminal papers (e.g. PMID 10911963), and to experimental evidence linking these processes to sarcopenia (e.g. PMID 31144432 and others). Also, figure 1 should take these concepts under consideration.
Minor:
-I would recommend proofreading for checking for minor typos.
-The authors should ensure that all the major statements are accompanied with proper references (e.g. lines 65-66).
-Present all the acronyms in their extended forms at the first occurrence (AMPK, PGC1a, TNFa, IL-6)
Author Response
Reviewer 1
Here, the authors reviewed the link between sarcopenic obesity (SOB) and a number of systemic/organ diseases. Their work is well organized, and focuses on novel aspects involving gut microbiota. Figures are appropriate and help the readers understand the key message of the review.
My only major concern is the total absence of any mention to "inflammaging" and "metaflammation" as cardinal manifestations of the aging phenotype. In particular, the authors are strongly encouraged to revise paragraph #2 introducing these concepts with proper citations to seminal papers (e.g. PMID 10911963), and to experimental evidence linking these processes to sarcopenia (e.g. PMID 31144432 and others). Also, figure 1 should take these concepts under consideration.
Response) Thank you for the comments. I edited the manuscript according to the comments. Paragraph #2 has more references that the reviewer suggested to add. Also, Figure1 has the concepts. The concept of “inflammaging” and “metaflammation” was described at line 71-76. Thank you again for the comments.
Minor:
-I would recommend proofreading for checking for minor typos.
Response) We checked it again. If there are still typos, please let us know it. Thank you.
-The authors should ensure that all the major statements are accompanied with proper references (e.g. lines 65-66)
Response) We have added the reference to line 70-71.
-Present all the acronyms in their extended forms at the first occurrence (AMPK, PGC1a, TNFa, IL-6)
Response) We added the full name of AMPK, PGC-1a, TNF-a and IL-6 on line 17-18, line 48-49, and line 70-71.
Reviewer 2 Report
In this article, the authors presented the association of sarcopenia obesity with other age-related diseases, its inducers and their crosstalk in a very interesting point of view. The manuscript would be improved if there is more exploitation regarding the IL-6 paradox in this condition. This theme is raised in line 149, however it should be more detailed. Regarding line 155, it is not clear how the lipid metabolism is improved in skeletal muscle. What is the proposed mechanism? Is it related with AMPK activation? Line 187-188, the definition should be inserted the first time that the authors mention autophagy. Line 326, remove the whole name of ACC. This abbreviation was inserted previously in line 288. Explain how AMPK stimulates lipid metabolism through the phosphorylation of ACC. For this protein, the phosphorylation inhibits the enzymatic activity. The paper is generally well written and presented. However, some English revision is needed and there are some sentences that are similar to each other and are presented in different sections of the review. I therefore recommend acceptance after minor revisions.Author Response
Reviewer 2
In this article, the authors presented the association of sarcopenia obesity with other age-related diseases, its inducers and their crosstalk in a very interesting point of view.
The manuscript would be improved if there is more exploitation regarding the IL-6 paradox in this condition.
This theme is raised in line 149, however it should be more detailed.
Response) We described more about IL-6 in line 157-160.
Regarding line 155(153), it is not clear how the lipid metabolism is improved in skeletal muscle. What is the proposed mechanism? Is it related with AMPK activation?
Response) We explained the mechanism in line 166-167.
Line 187-188, the definition should be inserted the first time that the authors mention autophagy
Response) We added the definition of autophagy in line 198-199.
Line 326, remove the whole name of ACC. This abbreviation was inserted previously in line 288.
Response) Thank you. It was removed in line 338.
Line 328, Explain how AMPK stimulates lipid metabolism through the phosphorylation of ACC. For this protein, the phosphorylation inhibits the enzymatic activity.
Response) Thank you for the comment. We described more about it at line 338-341
The paper is generally well written and presented. However, some English revision is needed and there are some sentences that are similar to each other and are presented in different sections of the review.
Response) Sorry about it. AMPK and PGC-1a signaling pathways are the main topic to be involved in SOB. Thus, the related explanation would be repeated. Thank you.
I therefore recommend acceptance after minor revisions.
Response) Thank you.